# Effect of Microsatellite Status and Pan-Immune-Inflammation Score on Pathological Response in Patients with Clinical Stage III Stomach Cancer Treated with Perioperative Chemotherapy

**DOI:** 10.3390/medicina59091625

**Published:** 2023-09-08

**Authors:** Ahmet Gulmez, Hatice Coskun, Tolga Koseci, Serdar Ata, Berna Bozkurt, Timucin Cil

**Affiliations:** 1Medical Oncology Department, Kisla Campus, Adana Baskent University, Adana 01120, Turkey; 2Adana State Hospital, Adana 01150, Turkey; drhaticecoskun@hotmail.com (H.C.); drserdarata@gmail.com (S.A.); berboz@hotmail.com (B.B.); drtimucincil@gmail.com (T.C.); 3Medical Oncology Department, Faculty of Medicine, Cukurova University, Adana 01380, Turkey; drtolgakoseci@gmail.com

**Keywords:** gastric cancer, microsatellite instability, neoadjuvant chemotherapy, pan-immune-inflammation score, pathological response

## Abstract

*Background and Objective*: This study evaluated the relationship between microsatellite status (MSI) and pan-immune-inflammation score (PIV) in tumor response to neoadjuvant chemotherapy (NAC) in patients with clinical stage III gastric cancer (cStage III GC). *Materials and Methods*: Microsatellite instability (MSI) status was evaluated based on pathology preparations. Pan-immune-inflammation score (PIV) was obtained from pre-treatment blood tests. The relationship of both parameters with pathological complete response (pCR) was evaluated. *Results*: A total of 104 patients were included in this study. All the patients were stage III GC patients receiving perioperative treatment. There were 13 patients in total who achieved a pCR response. While CNS was detected in 11 of the patients who achieved a pCR, the MSI status of the other two patients was unknown. No pCR was observed in any patient with MSI-H. According to the cut-off value for PIV, 25 (24%) patients were in the PIV-low (≤53.9) group, while 79 (76%) were in the PIV-high (>53.9) group. Based on univariate analysis, a higher PIV was associated with worse outcomes for pathological response, disease recurrence, and survival (*p* < 0.05). *Conclusions*: In patients with clinically stage III GC, the presence of MSI-H may predict no benefit from perioperative treatment. Conversely, a pre-treatment PIV score using specific cut-off values may provide a positive prediction of pathological response and survival.

## 1. Introduction

Gastric cancer is closely associated with mortality and is the third most common cause of cancer death worldwide [1]. Surgical procedures other than palliation are not the standard treatment option in the treatment of gastric cancer diagnosed in the metastatic stage. In contrast, if gastric cancer is detected at an early stage at diagnosis, lymph node dissection combined with gastrectomy is the main component of gastric cancer (GC) treatment. However, the prognosis is quite poor in patients with locally advanced and/or lymph node (LN)-positive GC due to the risk of recurrence. This poor clinical condition also leads to routine recommendations for pre-surgical systemic chemotherapy (NAC) for operable stage II and III GCs [2]. Previously, for most tumors, surgery was preferred unless metastatic disease was present and surgical resectability was possible. However, patients’ decreased performance after major surgery and consequently decreased tolerability of adjuvant systemic therapy was an important problem. For this reason, the neoadjuvant treatment approach started to gain importance in appropriate patients. This approach allowed clinicians to assess the aggressiveness of tumor behavior. In accordance with the recommendations of international oncology guidelines, NAC is recommended as a standard before surgery in patients with locally disseminated GC. However, the response of patients who were operated on after NAC to chemotherapy is variable. While some patients have a pathological response (pCR) after chemotherapy, some patients do not seem to have any response to chemotherapy. In a previously published phase II study, the pCR rate of the FLOT (docetaxel, oxaliplatin, leucovorin, and fluorouracil) protocol, which provides superiority to the standard treatment, remains below 20% [3]. This shows that there are still more than 80% of patients who did not obtain an adequate response with NAC. Increasing this response rate may be possible by examining tumor pathogenesis and developing treatment options. Therefore, it is extremely important to determine which patients achieve pCR. There are some parameters that are thought to be associated with pathological complete response after systemic therapy. One of these is known as microsatellite instability. Recently, conflicting results have emerged regarding the response to systemic chemotherapy in patients with high microsatellite instability (MSI-H), which is associated with sensitivity to immunotherapy [4,5,6]. Detection of MSI-H as a result of pathological examination of some cancer types is associated with good immunotherapy response. Although this marker has often been tested in metastatic disease, evaluation of the neoadjuvant response to immunotherapy seems likely in the near future. There are some ongoing clinical studies on this subject, but detailed studies with large numbers of patients are needed.

Inflammation and cancer are terms that have been evaluated for a long time and are closely related to each other. Demonstration of inflammation at the cellular level is possible with some parameters. Inflammation severity can be indirectly shown by some parameters obtained using cellular blood elements. These markers are generally named inflammatory prognostic markers. Another marker of response to chemotherapy is inflammatory prognostic markers (IPM). Unlike other more commonly known IPMs, the pan-immune-inflammation score (PIV) is a newer marker. The pan-immune-inflammation score is slightly different from other inflammatory prognostic markers. The other markers usually include the ratio of two of the blood elements to each other, while the PIV is calculated with a formulation that includes all the blood elements.

As a result of an extensive literature review, we found that both MSI and PIV are generally associated with treatment response in metastatic diseases. For this reason, we thought that a study that included examining the response after neoadjuvant systemic therapy in patients with locally advanced gastric cancer would contribute to the literature. Our aim in this study is to examine the relationship between MSI-H and PIV and tumor response to NAC and its prognostic significance in patients with clinical stage (cStage) III GC treated with NAC and surgery.

## 2. Materials and Methods

All patients with a radiological diagnosis of stage III GC at Adana State Hospital between January 2019 and January 2023 and treated with NAC followed by curative gastroectomy were included in this study. All patients included in this study were diagnosed with gastric adenocarcinoma. Patients with a diagnosis other than adenocarcinoma were not included in this study. GCs in squamous-cell carcinoma morphology (SCC), neuroendocrine tumor or neuroendocrine carcinoma, mixed-type GC, gastric lymphoma, gastrointestinal stromal tumor (GIST), and those treated with non-curative resection were not included in this study. All these patients with adenocarcinoma were classified according to the Laurén classification. According to this classification, most of the patients were classified as intestinal type; diffuse type was the second most common type, and mixed type was the least common type, which, according to this classification, were 59%, 34%, and 7%, respectively. In addition to these criteria, patients with missing pathology records were also excluded from this study. In all patients included in this study, a simultaneous biopsy was performed with esophagus-gastro-duodenoscopy to provide a histological diagnosis. In addition, thoracic–abdominal–pelvic computed tomography (TAP-CT) or positron emission tomography (PET) performed depending on physician preference were applied to determine tumor staging. In accordance with international oncology guidelines [7], the FLOT protocol was administered as NAC for 4 cycles to patients with clinical Stage III GC according to the International Association for Cancer Control (UICC)/American Cancer Committee (AJCC) TNM classification, 8th edition [8]. After 4 cycles of NAC, the pre-treatment imaging method was routinely reapplied to the patients, and then the patients were referred for surgery. After the postoperative pathological evaluation and wound healing, 4 cycles of adjuvant FLOT were applied to the patients. All patients underwent laparoscopic evaluation before curative surgery. After laparoscopy, patients without peritoneal metastases were operated on with gastrectomy and at least DII lymph node dissection. Tumor response to NAC was evaluated according to the degree of tumor regression (TRG) described in the study by Becker et al. According to this classification, patients were divided into three groups. A residual tumor of <10% was classified as Grade 1. If 10–50% residual tumor remained, it was classified as Grade 2. Detection of >50% tumor was classified as Grade 3. [9]. Pathological complete response was evaluated as the absence of residual tumor cells in both the primary tumor and lymph nodes. MSI status of pathology specimens was determined using a five-point panel of Bethesda markers (MONO-27, BAT-25, CAT-25, NR-24, BAT-26) [10]. Tumors with instability ≥2 from the five markers were classified as MSI-H. Those with one labile marker were classified as low microsatellite instability (MSI-L), while tumors with all five markers stable were classified as microsatellite stability (MSS). Four mismatch repair (MMR) proteins (MSH6, MSH2, PMS2, MLH1) were subjected to immunohistochemical (IHC) staining in tumor tissue sections. Loss of expression of a single protein or a dimeric pair (MSH2/MSH6 or MLH1/PMS2) suggests the presence of MMR deficiency (dMMR) [11]. As described above, this treatment is routinely administered in our oncology center to all patients with an indication for neoadjuvant chemotherapy. All patients in our study had an indication for neoadjuvant chemotherapy and were planned to receive this treatment. All patients in this study received FLOT chemotherapy as a neoadjuvant systemic treatment. None of the patients received a treatment regimen other than FLOT as neoadjuvant chemotherapy. In all of the patients in this study, this treatment was planned for a total of 4 cycles every 2 weeks before surgery. An additional 4 cycles of treatment were planned after surgery. Patients receiving neoadjuvant systemic therapy other than the FLOT regimen were not included in this study.

In this study, the results of peripheral blood tests taken on the day of the start of treatment were examined. Absolute neutrophil, lymphocyte, platelet, and monocyte counts were recorded. For the calculation of the pan-immune-inflammation score (PIV), the complete blood count parameters of the patients before treatment were used. The pan-immune-inflammation score (PIV) was calculated as neutrophil count × platelet count × monocyte count/lymphocyte count. The absolute value of this parameter was calculated and noted for each patient and used for analysis. This study was approved by the Local Ethics Committee of Adana State Hospital (06 April 2023, 2428). Patient data were collected according to ethical principles for medical research involving human subjects accepted in the Declaration of Helsinki. Since our study was a retrospective study in which only hospital records were used, an informed consent form was not obtained from the patients after obtaining the approval of the ethics committee.

## 3. Statistical Analysis

Analysis of clinical data was performed retrospectively. Overall survival (OS) was calculated as the time from diagnosis of the disease to death or last follow-up. Disease-free survival (DFS) was calculated from the time of primary diagnosis to recurrence, death, or last follow-up. The conformity of continuous variables to normal distribution was examined by visual (histogram) and analytical methods (Kolmogorov–Smirnov). In descriptive analysis, mean and standard deviations were used for normally distributed variables, while median and interquartile range values were given for non-normally distributed variables. Independent groups *t*-test was used to compare normally distributed numerical variables. Chi-squared or Fisher’s exact tests were used to evaluate the relationship between PIV and clinical parameters. The relationship between patient clinical characteristics and survival times was evaluated using Kaplan–Meier survival analysis and log-rank test. The diagnostic determinants of the pan-immune-inflammation value (PIV) in predicting pCR were analyzed by ROC curve analysis. For the evaluation of the analyses, SPSS version 20.0 software was used, and a *p*-value of <0.05 was accepted for statistical significance.

## 4. Results

In this study, 104 patients who received NAC were evaluated. The data of these patients were analyzed statistically.

The majority of patients included in this study were men. Approximately 80% of the patients were male and 20% were female. The mean age of the patients in this study was 60.5. The ECOG performances of the patients included in this study were also evaluated. Approximately 80% of the patients whose ECOG performance was evaluated retrospectively from hospital records were found to be ECOG:0, while 20% were found to be ECOG:1. Patients with ECOG:2 and above were not included in this study. The patients were also evaluated for CERB-B2. While CERB-B2 was negative in approximately half of the patients, about 3% of the patients were positive. The remaining patients, which comprised approximately half of the patients in this study, did not have a CERB-B2 evaluation. Patients in this study were also classified according to tumor location. A small proportion (25%) of the patients were of gastro-esophageal junction origin, while 75% of them were of gastric origin. The patients were also evaluated according to the TNM classification. These patients were classified radiologically in terms of T score. While there were no T1 tumors among these patients, T2 tumors were detected in 10% of patients, T3 tumors in 70%, and T4 tumors in 20% of patients. While there were 3 patients who were evaluated as lymph node negative, the remaining 101 patients had clinical lymph node positivity. Patients were grouped according to the Laurén classification. About 60% of patients were evaluated as intestinal type, while 34% of patients were classified as diffuse-type gastric cancer. Apart from this, the remaining 7% of the patients were classified as mixed type. Tumor marker elevations at the time of diagnosis were also evaluated. Carcinoembryonic antigen (CEA) was above the value considered normal at the time of diagnosis in 70% of the patients. Similarly, the CA 19-9 level was also evaluated. This marker was elevated in 60% of all patients in this study. The clinical and demographic information of the patients are shown in Table 1. A four-cycle FLOT regimen was planned as neoadjuvant therapy for all patients in this study. It was determined that most of the patients (96%) completed the treatment as planned. Four patients could not complete four cycles of planned neoadjuvant therapy for various reasons. A total of 10 (9%) patients required dose reduction due to hematological toxicities.

All patients in this study were prepared for surgery by radiological evaluation after neoadjuvant systemic therapy. Total gastrectomy was performed as a surgical procedure in the vast majority of patients. Total gastrectomy was performed in 85 patients (82%), while subtotal gastrectomy was performed in 19 (18%) patients. After surgery, both the pCR and the degree of response were evaluated in patients without a pathological response. As a result of the evaluation, pCR was observed in 13 (12.5%) of all patients included in this study. There was no pathological response in the remaining 91 patients, and <10% vital tumor cells were detected in 11 of these patients. It was observed that 22 of them had 10–50% vital tumor cells, and 58 of them had >50% vital tumor cells. The microsatellite status of the patients who achieved a pathological response was also evaluated. None of the patients in the MSI-H group had pCR. On the other hand, pCR was detected in 11 of the patients in the MSS group. It was observed that MSI status was not specified in the pathology report of two patients who achieved a pCR. This pathological response assessment is shown in Table 2. It was obtained by ROC curve analysis using parameters evaluated before neoadjuvant systemic therapy. This analysis is illustrated in Figure 1 below. To predict pCR, a PIV value of 53.9 was determined as the cut-off value with a sensitivity and specificity of 84.6%. Univariate Cox regression analysis showed that pretreatment PIV score is important for predicting pathological response after neoadjuvant systemic therapy. In 11 of 13 patients who achieved a pCR, the PIV value obtained as a result of the laboratory parameters measured before the treatment was found to be below 53.9. However, there were two patients in total who had a PIV score above this cut-off value and achieved a pCR. The difference between these groups was statistically significant (*p* < 0.05). Similarly, the relationship between PIV and relapse status and overall survival was also evaluated. Similar to pCR, there was a statistically significant correlation between PIV score in both recurrence and survival. While 11 (44%) of the patients with a PIV score below the cut-off value of 53.9 relapsed, 52 (66.7%) of the patients with a PIV score above 53.9 relapsed (*p* < 0.05). The OS of the patients whose PIV score was below the cut-off value of 53.9 was 40.8 ± 3.9 months, while the overall survival of the patients above this value was 33.6 ± 2.8. These survival times also differed statistically significantly (*p* < 0.05). Information showing the relationship between PIV score and pathological response, recurrence rates, and survival is given in Table 3.

## 5. Discussion

This study showed that microsatellite instability is significantly associated with unresponsiveness to neoadjuvant chemotherapy in patients with cStage III GC. In our study, a total of 11 (10%) patients were found to have MSI-H. In a large-numbered meta-analysis in the literature, it was found to be around 9%, similar to this study [12]. A worse histological response of MSI-H to neoadjuvant chemotherapy was previously observed, which was not statistically significant (*p* > 0.05) [4,5,6]. In another study, Hashimoto et al. noted the loss of MLH-1 expression as an indicator of worse histological regression after neoadjuvant chemotherapy, evaluated according to the Japanese GC Society criteria [5].

Another important study for MSI-H stomach cancer is the MAGIC study. This is important because a second analysis of this study investigated the efficacy of treatment with MSI status in gastric cancer receiving perioperative chemotherapy. In this study, the rate of tumors with MSI-H/MMR protein deficiency was reported to be 7% [4]. And these patients with MSI-H had better results with surgery alone than in patients with MSS. In contrast, the median OS in patients receiving perioperative chemotherapy was shorter in those with MSI-H tumors. When comparing patients with MSI-H and patients with MSS, OS was 9.6 vs. 19.5 months, respectively. This difference was also statistically significant. However, there are some factors that should be known about the MAGIC trial. Anthracycline-based treatments were also used in the perioperative period in this study. Treatment toxicities possibly related to these treatments may also be associated with this poor survival. Regarding adjuvant therapy, Choi conducted a second analysis of the CLASSIC study. This analysis compared surgery alone with surgery and adjuvant chemotherapy. Choi reported that MSI-H cancer patients had better recurrence-free survival after surgery alone compared to non-MSI-H cancer patients. No benefit from adjuvant chemotherapy was observed in MSI-H cancer patients in this analysis [13]. In another retrospective cohort study [14] with a large number of patients, Kim et al. showed that patients with stage II/III MSI-H tumors did not benefit from adjuvant chemotherapy. Even the deleterious effect of adjuvant chemotherapy was observed in some subsets of patients (those with stage III disease, undifferentiated histology, and diffuse-type tumors). However, after the German FLOT4 study [2], perioperative management was changed to a combination of taxane and platinum rather than standard anthracycline-based treatments. Therefore, studies evaluating MSI status in patients treated with perioperative FLOT are needed. This trial that we have completed is important. In our study, there was no patient who achieved a pCR with FLOT treatment. However, there is no detrimental effect on survival in patients receiving chemotherapy. With a single-center and retrospective study, it is impossible to say that MSI-H tumors did not benefit from NAC. However, it is known from recent studies that immune checkpoint inhibitor therapy is a new hope for MSI-H GCs [15]. It is well known that immunotherapy is efficient for MSI-H metastatic or refractory tumors [16]. However, there is insufficient evidence to replace perioperative or adjuvant chemotherapy for cStage III MSI-H GC. Zheng et al. reported a high rate of pCR (83.3%) in MSI-H gastrointestinal tumors treated with neoadjuvant immunotherapy. However, it should be noted that this article is a case series with only six patients [17]. Therefore, future clinical trials for cStage III MSI-H GC may explore taxane-based chemotherapy regimens ± immunotherapy.

Inflammation is vital to innate immunity. This mechanism performs immune surveillance. In this way, it helps to protect the host from external attacks [18,19]. The inflammatory response of the innate immune system is the body’s first line of defense against carcinogens. As a result of the malfunctions that may occur in this protective system, the development and progression of cancer may occur. In contrast, uncontrolled inflammation may have an adverse effect on cancer development due to several mechanisms, including DNA damage by proinflammatory cytokines and chemokines and an increased risk of genomic alterations and instability [20]. Prognostic scores obtained from peripheral blood count items, which are accepted as an indirect measure of inflammatory burden in cancer, have recently started to attract increasing attention from researchers [21,22]. Numerous studies have been conducted on both the platelet-to-lymphocyte ratio (PLR) and the neutrophil-to-lymphocyte ratio (NLR). However, only two of the blood elements were used in these markers. Because of these insufficient inflammation measurement values, a new inflammation scale called the pan-immune-inflammation value (PIV) has been developed. An equation including monocytes, platelets, and neutrophil levels along with lymphocytes is used in the calculation of the PIV [23]. There are many studies that have previously examined the relationship between PIV and survival. These studies were evaluated in a comprehensive meta-analysis published recently [24]. In this context, a negative correlation between survival and PIV levels was observed in the pooled analysis of thousands of patients. In patients with high PIV values, it has been shown to be a consistently consistent negative prognostic factor in a variety of clinical scenarios, including whether or not metastatic disease is present and in patients treated with targeted therapy or immunotherapy. While there was a pCR in 44% of the patients with PIV ≤53.9, only 2.5% of the patients with a PIV ≤53.9 had a pCR. The difference between these two groups is statistically significant.

PIV is a marker that has become increasingly important recently and has been studied in almost every cancer type. It was also recently evaluated in a meta-analysis, this time involving colorectal cancer patients [25]. This study included 1879 colon cancer patients. Similar to the previous studies, a significant relationship was found between PIV and both overall survival and disease-free survival in this study. This study, which we have completed, also overlaps with the results of the two different meta-analyses mentioned above.

This study also has some limitations. First, this study was both a retrospective and a single-center study. Well-designed prospective studies are needed to confirm our findings. Second, not all of our patients had MSI results. Since it was a retrospective study, it could not be re-evaluated. Despite all these limitations, we think that our study will contribute to the literature as it is an original study.

In conclusion, the current treatment in clinical diagnosis of stage III GC is the surgical approach combined with perioperative FLOT therapy. One of the most important goals of neoadjuvant systemic chemotherapy is to achieve a pathological complete response in surgery to be performed after completion of treatment. It is extremely important to distinguish patients with whom we can obtain this complete response from others. Similarly, more effective predictive markers are needed to protect patients in whom neoadjuvant therapy is not effective from the toxicities of systemic therapy. Two important markers that we evaluated in our study provided a prediction of pathological response. Our study showed that the group with MSI-H had a low pCR rate with perioperative systemic therapy, whereas patients with a low PIV score had a high pCR rate.

## Figures and Tables

**Figure 1 medicina-59-01625-f001:**
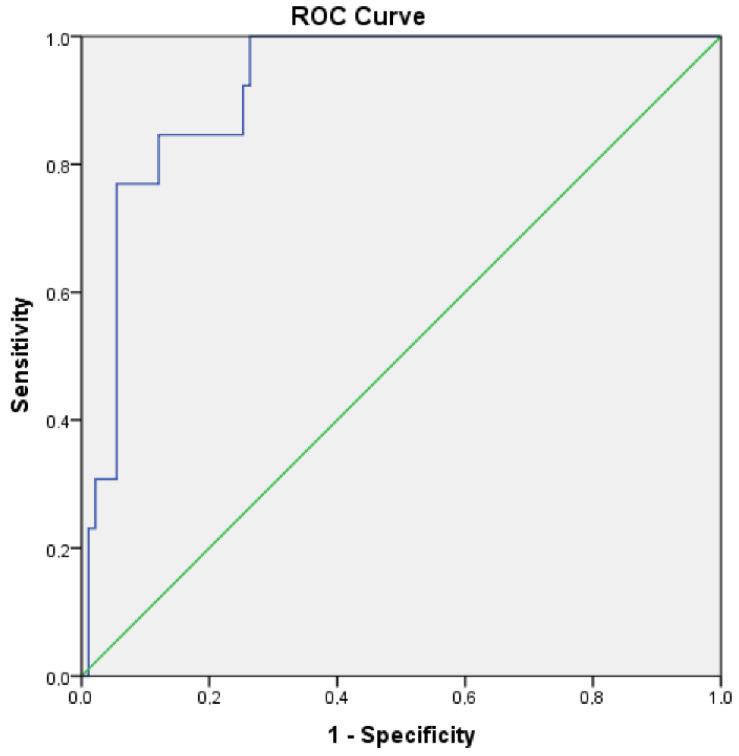
ROC curve analysis.

**Table 1 medicina-59-01625-t001:** Patient demographics and clinical characteristics.

	Number of Patients (n = 104)	(%)
Sex		
Male	81	78
Female	23	22
Age, Years		
Median	60.5	
Range	22–86	
ECOG		
0	80	77
1	24	23
CERB-B2		
Negative (IHC 0, IHC 1+, IHC2+ and FISH neg.)	54	52
Positive (IHC 2+ and FISH positive or IHC 3+)	3	3
Not Determine	47	45
Location		
Gastro-Esophageal Junction	26	25
Gastric Cancer	78	75
T-stage		
T1	-	-
T2	10	10
T3	73	70
T4	21	20
Nodal Stage		
cN−	3	3
cN+	101	97
Laurén Classification		
Intestinal type	61	59
Diffuse type	35	34
Mixed type	8	7
CEA Levels at Diagnosis		
High	72	70
Normal	15	14
Unknown	17	16
Ca19.9 Levels at Diagnosis		
High	62	60
Normal	23	22
Unknown	19	18

ECOG—Eastern Cooperative Oncology Group; CERB-B2—humanized epidermal growth factor receptor 2; CEA—carcinoembryonic antigen.

**Table 2 medicina-59-01625-t002:** Histological regression after neoadjuvant treatment.

Becker Classification	Number of Patients (n=104)	%
Complete Response (IA)	13	12.5
<10% Vital Tumor (IB)	11	10.5
10–50% Vital Tumor	22	21
>50% Vital Tumor	58	56
Complete Response (IA)	**MSI-H**	**MSS**
13 patients (%12.5)	-	11(%19)
ypN Stage		
pN−	-	23/59 (%39)
pN+	11	36/59 (%61)
ypT Stage		
pN−	-	14/59 (%24)
pN+	11	45/59 (%76)

MSI-H—microsatellite instable; MSS—microsatellite stable.

**Table 3 medicina-59-01625-t003:** Relationship between PIV and pathological response, survival, and recurrence.

	PIV ≤ 53.9 n = 25 (%)	PIV > 53.9 n = 79 (%)	*p*-Value
Pathological Complete Response			
Yes	11 (44.0)	2 (2.5)	
No	14 (56.0)	77 (97.5)	*p* < 0.05
Recurrence			
Yes	11 (44.0)	52 (66.7)	*p* < 0.05
No	14 (56.0)	26 (33.3)	
Overall Survival			
(month ± std)	40.8 ± 3.9	33.6 ± 2.8	*p* < 0.05

PIV: pan-immune-inflammation value.

## Data Availability

The data and materials used in this study are available upon reasonable request from the corresponding author. Restrictions may apply to the availability of certain data sets due to privacy or ethical restrictions.

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
