# Peer review of "Effect of Microsatellite Status and Pan-Immune-Inflammation Score on Pathological Response in Patients with Clinical Stage III Stomach Cancer Treated with Perioperative Chemotherapy"

_medicina, 2023, doi:10.3390/medicina59091625_

Round 1

Reviewer 1 Report

1. How is PIV calculated? It was not clearly stated in the study methodology.

2. Perioperative Chemotherapy should specify the treatment time and plan, how to solve the error problem caused by different chemotherapy schemes?

3. The research results are recommended to be displayed in the form of figures or tables.

4. There were no general clinical features for 104 patients. Are the types of pathology different in all patients? Different pathological types may cause errors in the research results, how can they be avoided?

Minor editing of English language required.

Author Response

Dear Reviewer-1;

First of all, thank you very much for reviewing my article and finding your comments valuable.

In response to your valuable question; I offer you my modest answer.

  1. How is PIV calculated? It was not clearly stated in the study methodology: I have corrected this definition. I have stated in detail in the material and method section
  2. Perioperative Chemotherapy should specify the treatment time and plan, how to solve the error problem caused by different chemotherapy schemes? I also fixed this confusion in the material and method section. All patients had received the same systemic neoadjuvant chemotherapy. This was written in the material and method part, with a clear correction.
  3. The research results are recommended to be displayed in the form of figures or tables. First of all, I apologize for this mistake. Both table and figure documents were available, but I did not put these documents in the main document. I have uploaded these documents as a separate file. I fixed this error. Both table and figure are included in the main document.
  4. There were no general clinical features for 104 patients. Are the types of pathology different in all patients? Different pathological types may cause errors in the research results, how can they be avoided? Explanations about this are included in the article. All of the patients were gastric adenocarcinoma patients. Patients with other pathological subtypes were excluded from the study. Necessary information has been added to the main article file.

Thank you very much

Kind Regards

Reviewer 2 Report

The authors have very well written the manuscript with a sound rationale, descriptive material and methods and well analyzed and discussed results. Although the results arent very impactful as also pointed by the authors in the manuscript, I think a well put results as this should be published. However, one of my questions is why did the authors choose to do their study pre-surgery chemotherapy instead of post operative chemotherapy which is said to more efficient to reduce tumour recurrence.

Author Response

Dear Reviewer-2;

First of all, thank you very much for reviewing my article and finding your comments valuable.

In response to your valuable question; I offer you my modest answer.

As is known, neoadjuvant chemotherapy approaches including systemic chemotherapy, immunotherapy and/or radiotherapy before surgery have become the standard treatment for most cancers. This approach has gained importance due to the morbidity that may prevent patients from receiving systemic treatment after a major surgery and the low tolerability of adjuvant treatment. In addition, preoperative systemic therapy is very important for the clinician to test the behavior pattern of the tumor. In addition, especially recently, pathologic complete response after neoadjuvant treatment has changed the principles of oncologic treatment. Long-term outcomes of patients who achieved pathologic complete response have improved significantly compared to patients who did not achieve pathologic complete response. In addition, internationally recognized guidelines such as NCCN and ESMO report that systemic therapy is the standard of care for gastric cancer patients with positive lymph nodes and/or radiologically >T2. All patients in our study were diagnosed with stage III gastric cancer. Therefore, systemic chemotherapy was used as standard for all patients in our study.

Thank you very much

Kind Regards